# Insecticidal Effect of Zinc Oxide Nanoparticles against *Spodoptera frugiperda* under Laboratory Conditions

**DOI:** 10.3390/insects12111017

**Published:** 2021-11-11

**Authors:** Sarayut Pittarate, Julius Rajula, Afroja Rahman, Perumal Vivekanandhan, Malee Thungrabeab, Supamit Mekchay, Patcharin Krutmuang

**Affiliations:** 1Department of Entomology and Plant Pathology, Faculty of Agriculture, Chiang Mai University, Chiang Mai 50200, Thailand; sarayut_pit@cmu.ac.th (S.P.); rajula_j@cmu.ac.th (J.R.); afroja_r@cmu.ac.th (A.R.); 2Society for Research and Initiatives for Sustainable Technologies and Institutions, Grambharti, Amarapur Rd, Gujarat 382735, India; Mosqvk@gmail.com; 3Agricultural Technology Research Institute, Rajamangala University of Technology Lanna, Lampang 52000, Thailand; mthungrabeab@rmutl.ac.th; 4Department of Animal and Aquatic Sciences, Faculty of Agriculture, Chiang Mai University, Chiang Mai 50200, Thailand; supamit.m@cmu.ac.th; 5Innovative Agriculture Research Center, Faculty of Agriculture, Chiang Mai University, Chiang Mai 50200, Thailand

**Keywords:** malformations, nanotechnology, *Spodoptera frugiperda*, zinc oxide nanoparticles

## Abstract

**Simple Summary:**

Fall armyworm has devastated several crops around the world, especially maize that is widely grown and utilized globally. Also, it has been known to cause a lot of damage in rice fields. However, controlling this pest has been a challenge to farmers due to its ability to reproduce faster and its development of resistance to synthetic chemicals, among other factors. Moreover, synthetic chemicals are a threat to the environment and humanity. For these reasons, we are constantly looking for safer yet effective means of controlling this pest, and nanotechnology comes in handy. Zinc Oxide nanoparticles have proved to be efficacious to several insect pests, of which some are in the same genus as *Spodoptera frugiperda*. This study aimed to find out the insecticidal effects of ZnO nanoparticles on *S. frugiperda* under laboratory conditions. We observed body deformations, reduced fecundity, reduced oviposition, and mortality when insects were fed on food treated with several concentrations of ZnO nanoparticles, yet the ones fed on control were normal in all the aspects. Therefore, we recommend ZnO nanoparticles for further studies with the aim of using them as an alternative control agent against fall armyworm under field conditions.

**Abstract:**

Fall armyworm *Spodoptera frugiperda* is a major pest of corn, rice, and sorghum among other crops usually controlled using synthetic or biological insecticides. Currently, the new invention of nanotechnology is taking root in the agricultural industry as an alternative source of pest management that is target-specific, safe, and efficient. This study sought to determine the efficacy of commercial Zinc Oxide (ZnO) nanoparticles (NPs) towards *S. frugiperda* under laboratory conditions. ZnO NPs were diluted into different concentrations (100–500 ppm), where the baby corn used to feed the *S. frugiperda* larvae was dipped. The development of the insect feeding on food dipped in ZnO solution was significantly (*p* < 0.05) affected, and the number of days that the insect took to complete its life cycle had a significant difference compared to the control. There was a significant difference in the adults’ emergence in all the concentrations of ZnO NPs compared to the control, with over 90% of the eggs successfully going through the life cycle until adult emergence. Additionally, several body malformations were observed throughout the lifecycle of the insect. Also, the fecundity of the females was greatly affected. The findings of this study suggest the possibility of exploitation of ZnO nanoparticles not only to manage *S. frugiperda* but to significantly reduce their population in the ecosystem through body deformations, reduced fecundity, reduced oviposition, and hatchability of eggs. It will be a valuable tool in integrated pest management regimens.

## 1. Introduction

*Spodoptera frugiperda* (Smith, 1797) otherwise, known as Fall armyworm has been a cause of menace to the maize crop for a long period mostly in the tropics and subtropics of America. That notwithstanding, the insect invades a number of other crops in the gramineous family including sorghum, millet, rice, Bermuda grass among others, making it a serious pest in the agricultural industry [1]. Recently, the insect has been reported in various African and Asian countries including Thailand. It has been estimated that fall armyworm has instigated a loss of between 2 and 6 billion U.S. dollars in Europe and Africa combined annually [2]. In America, the losses caused are so enormous, and controlling the insect has been a daunting task. For a long time, chemical pesticides have been employed in the control of this pest in America, Europe, and now the newly invaded regions in Africa and Asia. However, the increased use of chemical insecticides causes environmental contamination, bioaccumulation, and human toxicity. Also, prolonged application of the compounds found in these chemicals accounts for resistance in several insect pests in agriculture including fall armyworm [3]. That notwithstanding, for chemical pesticides to work effectively, timing must be accurate for proper management of the same.

Globally, scientists and governments are shifting from chemical-based agriculture to green agriculture in order to ensure safety of the environment, animals, and human beings. This they tend to achieve by using plant extracts, entomopathogenic microorganisms, and nanoparticles to control plant pests and diseases [4]. Also, plants have also risen to the task of defending themselves from herbivory attack by producing volatile organic compounds (VOCs) that have been observed to reduce the insect invasion by up to 90%. This has been observed in maize to control *Ostrinia nubilalis*; the stored products pest *Sitophilus granaries* and many other insects, especially the ones that cause injury to their host as the VOCs are triggered by injury. Luckily, this is not dependent on their taxonomic affinities. More is being explored by scientists in this area so that the volatiles can be commercially availed for management of plant insect pests [5,6,7]. Additionally, a variety of metal particles such as zinc (Zn), silver (Ag), copper (Cu), silica (Si), gold (Au), aluminum (Al), and metallic oxides such as Titanium and Zinc oxides are being synthesized for use in controlling plant pests and diseases [8].

Nanoparticles are characterized by atomic or molecular size less than 100 nm [9]. Slowly but steadily, they have gained popularity due to their fascinating properties compared to their competitors in terms of applications [10]. Their high solubility in water and stability of the formulations are outstanding as compared to other pesticides [11,12]. Additionally, nanoparticles have proved to be efficacious against plant pathogens, weeds, and insect pests. Also, they have been added to the formulations of insecticides and insect repellents. Fortunately, the nanoparticles do not pose health hazards to the environment and public health compared to the traditional chemical pesticides [13,14,15]. However, the detailed mechanisms of nano-particles have not been understood to a convincing conclusion and therefore, need in-depth examination or experimentations of their interaction with the biological systems before designing consumable products [9]. Generally, nanoparticles are known to permeate through plant cells hence they become nanocarriers making them efficient in targeting the pest [12]. Zinc oxide nanoparticles (ZnO NPs) are among the significant metal oxides popular due to their chemical and physical peculiarity [16,17]. Notably, ZnO has proved to possess great potential in the biosynthesis of nanoparticles for clinical purposes compared to other oxides. Moreover, several studies have confirmed that ZnO nanoparticles can be synthesized using various plant extracts such as *Hibiscus rosasinensi* and *Cassia auriculata* [18]. That notwithstanding, ZnO nanoparticles synthesized using plant extracts as reducing agents have proved to have sturdy antimicrobial efficacy against *Aspergillus* spp., *Pseudomonas aeruginosa*, *Staphylococcus aureus*, and *Klebsiella pneumonia* [19,20]. This study, therefore, sought to determine the efficacy of commercial ZnO nanoparticles towards *Spodoptera frugiperda* under laboratory conditions. Additionally, the study elucidated the developmental interference after the consumption of food dipped into ZnO nanoparticles solutions of different concentrations.

## 2. Materials and Methods

### 2.1. Area of the Study

The experiments were conducted at the Laboratory of Insect Pathology, Department of Entomology and Plant Pathology, Faculty of Agriculture, Chiang Mai University, Muang District, Chiang Mai province, Thailand. The region is a plain with an altitude of 311 m above sea level at a longitude of 98.97° E and a latitude of 18.77° N. Due to the invasive nature of the fall armyworm, all observations were recorded under quarantine facilities.

### 2.2. Fall Armyworm Rearing

Larvae, pupae, and adults of fall armyworm (FAW) used in this study are the laboratory strain maintained at the Insect Pathology Laboratory, Department of Entomology and Plant Pathology, Faculty of Agriculture, Chiang Mai University, Thailand. This colony originated from a wild strain larvae captured from the cornfield at Mae-Hia Agricultural Training and Research Center, Faculty of Agriculture, Chiang Mai University during the corn/maize growing season and had been maintained in plastic containers (19 cm in width by 27 cm in length by 8 cm in height) and fed on baby corn under laboratory room conditions at 26 ± 1 °C with a photoperiod of 12:12 h (dark:light) and 70 ± 10% humidity.

### 2.3. Source of Zinc Oxide Nanoparticles

Extra pure zinc oxide (25–50 nm) nanoparticles were purchased from Green Nanotechnology (Chiangmai, Thailand). Complete characterization of the particles is available at the company and can be accessed through their website http://www.nano.kmitl.ac.th (accessed on 4 November 2021). Naturally, the aqueous suspensions of nanoparticles are stable but usually, they suffer aggregation in water; hence, the suspensions were vortexed for 10–20 min before use in the treatments [3].

### 2.4. Bioassay of Fall Armyworm Reared on Different Concentration ZnO NPs

Due to the invasive status of this insect pest in Thailand, the egg batches used in this study were derived from the first generation of Insect Pathology Laboratory (IPL) stocks of FAW females captured from the cornfield at Mae-Hia Agricultural Training and Research Center, Faculty of Agriculture, Chiang Mai University as previously mentioned.

The egg batches that were oviposited were maintained under controlled conditions, and upon hatching, the neonate larvae were individually placed in plastic containers measuring (sauce cup 6 cm × 3 cm; 3 oz). Various concentrations were prepared by diluting the original ZnO nanoparticles using distilled water to achieve 100, 200, 300, 400, and 500 ppm respectively. Thereafter, the achieved concentrations were administered orally to the neonate larvae of fall armyworm by feeding them on baby corn weighing approximately 1–3 g dipped in the various ZnO NPs concentrations. This was done every 3 days until they reached the pre-pupal stage. On the other hand, the control group were fed on baby corn that were not charged with nanoparticles [9]. Each day, the development and survival of larvae were observed for each treatment. Larval stages were L1, L2, L3, L4, L5, and L6. Pupal weight, width, length, and deformed were recorded from the surviving insects, and sex determination was conducted at the pupal stage by observing under Zeiss Stemi 508 Compound microscope [21]. Observations of insect biological parameters and development including duration of instars, fertility, deformities, and longevity of individuals were recorded daily. The duration of instars was determined by the number of days the larvae takes from one instar to the next from neonate larvae to the sixth instar. Fertility was determined by the number of eggs oviposited by the females and their hatchability and deformities was determined by observing the physical appearance of the larvae, pupa, and adults. Each treatment had four replications of twenty larvae each and the completely randomized block design was used.

### 2.5. Oviposition Preference

To determine which cumulative ZnO NPs’ different concentrations in adults are preferred for oviposition. Oviposition substrate was used to build the resulting eggs corresponding to the treatment food they were offered. After adult emergence, a total of 30 FAW pairs were used for this experiment, with one virgin pair (one male and one female) released at the center of a plastic container covered with plastic caps (drinking cups; 11.5 cm in width × 14.5 cm in length; 300 oz.), for a total of five replications. Female adults were allowed to oviposit for 7 days and fed with a 10% honey solution provided in small plastic boxes. The source of food used to nourish adults was being replaced at 2–3-day intervals. Folded pieces of paper (4 cm in width by 10 cm length) were hung in the rearing cage for egg-laying. Egg masses, egg numbers, and percent egg hatching were recorded daily until the death of each individual [22].

### 2.6. Statistical Analysis

Standard procedures were followed to record the data. The data collected was analyzed statistically using a one-way analysis of variance (ANOVA) with Ducan’s posthoc test to assess the oviposition preference and pupal weight, length, and width of FAW among different concentrations of ZnO NPs. The treatment means were compared using Tukey’s for their significance at the 0.05% probability level. The FAW egg masses and number of eggs recorded in the different concentrations of ZnO NPs were compared by paired T-test. Differences were considered significant at *p* < 0.05. The statistical analyses were conducted using the IBM SPSS Statistical Software package version 23.0 ( IBM Corp, Armonk, NY, USA, 2015).

## 3. Results

### 3.1. The Development of FAW Fed on Baby Corn Dipped in Different Concentrations of ZnO Nanoparticles

When the insect was fed on food dipped in ZnO nanoparticles, the mortality rate observed was not significant; however, there was a significant difference in the development of the insect from larval state to adults. The results obtained demonstrate the insecticidal activity of ZnO nanoparticles when orally ingested by the insect under a controlled environment in the assumption that no factors interfere with the outcome. The larvae feeding on food dipped in ZnO NPs solution were affected and the number of days taken to complete the lifecycle had a significant difference compared to the control. Also, a significant difference was observed in the emergence of adults compared to the control that had over 90% of the eggs successfully going through the cycle till adult emergence (Table 1, Table 2 and Table 3). When the pupal weights were taken, we observed significant differences (*p* < 0.05). The pupa fed on ZnO NPs treated baby corn weighed much less than the pupa from the control.

### 3.2. ZnO NPs Induced Morphological Changes in the Insect

Exposure to ZnO NPs caused body deformities in all stages of the life cycle from larvae to adults. Figure 1, Figure 2, Figure 3 and Figure 4 clearly show the body malformations instigated after the ingestion of baby corn dipped in ZnO NPs. Each figure clearly shows the typical morphology of larvae fed on control and the malformations that are caused due to the ingestion of food dipped in ZnO NPs. Compared to the control, there were significant differences (*p* < 0.05) in oviposition, larval development, pupal development, and adult emergence. The pupal weight, length, and width were taken, and it was observed that there was a significant difference in the weight of all the other treatments compared to the control. Also, the pupal length had a significant difference when the control was compared to the different concentrations of ZnO NPs. However, the pupal width had no significant difference.

### 3.3. Mortality of Spodoptera frugiperda Fed on Baby Corn Dipped in ZnO NPs

Our unpublished data showed mortality of about 40% of larvae after 10 days of feeding on baby corn dipped in ZnO NPs solution.

### 3.4. Effect of ZnO NPs on Fecundity and Fertility of Spodoptera frugiperda

The effect of ZnO NPs on female fecundity and fertility is presented in Table 4 and Figure 5. There were significant differences (*p* < 0.05) in the number of females that were able to oviposit and the ones that did not oviposit due to ZnO NPs treatment. For example, in Table 4, out of 22 females who fed on food dipped in ZnO NPs, only 7 (31.8%) proved to be reproductive, while over 74% of the females who fed on control were oviposited. That notwithstanding, the number of eggs oviposited by the latter was much more than those oviposited by the former. Moreover, the hatchability of the eggs laid by the females who ingested ZnO NPs ranged between 1.92 to 7.64%, while the eggs oviposited by the control hatched 96.4%. Additionally in Figure 5, it was observed that the eggs that were oviposited by females fed on baby corn dipped in ZnO NPs solution were not only a few by only a paltry hatched.

Moreover, the hatchability of the eggs laid by the females that ingested ZnO NPs ranged between 1.92 to 7.64% while the eggs oviposited by the control hatched 96.4%.

### 3.5. Effects of ZnO NPs on the Longevity of Spodoptera frugiperda

When treated with different concentrations of ZnO NPs, there was a significant difference (*p* < 0.05) in larval, pupal, male, and female adult longevity. The females who fed on control lasted 13 days, while the males lasted over nine and half days, which was way longer than the rest who ingested ZnO NPs (Table 1).

## 4. Discussion

This study aimed to determine the insecticidal effect of ZnO on the development of fall armyworm from neonate larvae to adults. *S. frugiperda* has majorly been managed using synthetic pesticides and a few biopesticides for the longest time that it has been ravaging crops. However, the use of these synthetic pesticides has damaged the environment and caused resistance to insecticides [23,24]. It poses a significant challenge in controlling this insect and hence the need to come up with a more subtle but effective means of controlling *S. frugiperda* that will not be dangerous to the environment and at the same time does not cause resistance to the insect [25]. In this regard, research is being conducted on alternatives that can effectively manage this insect and are also safe for the environment. ZnO NPs have proved to be a promising source of safe insecticides yet effective in controlling *S. frugiperda*. Also, it has been observed that it can manage several other insect pests at environmentally friendly and low dosages of about 700 mg mL^−1^ had the highest insect mortality effect [24]. Moreover, ZnO NPs were combined with Thiamethoxam and enhanced its efficacy in the management of *Spodoptera litura* [22]. The development of *S. frugiperda* was significantly jeopardized due to their exposure to ZnO NPs as displayed in Table 1. These observations corroborate other findings before with different insects such as rice weevil *Sitophilus oryzae* and *Trialeurodes vaporariorum* where mortality was recorded due to exposure to ZnO NPs [26,27]. Also, when leaves infested by *Aphis nerii* were dipped into various concentrations of ZnO NPs, there was a significant effect on their development [24]. Studies have been carried out on the insecticidal effects of silver nanoparticles and have shown that they can manage the insects that damage crops such as *Spodoptera littoralis* [28]. Another study using biosynthesized AgNPs showed larvicidal toxicity against *Anopheles stephensi*, *Aedes aegypti*, and *Culex quinquefasciatus* [29]. In addition, green copper nanoparticles when tested against *Aedes aegypti*, *Culex quinquefasciatus,* and *Anopheles stephensi* showed impressive results when measured using LC_50_ and LC_90_ [30].

It has been established that when there are body malformations in the insect population, the population is greatly reduced beneath economic levels and that is exactly what we observed in this experiment. There were several body deformations observed throughout the lifecycle of the insect as depicted in Figure 1, Figure 2, Figure 3 and Figure 4. Causing deformity in the insect pest population in the integrated pest management program is one of the important aspects in controlling pest population below the economic damage level. Apart from ZnO NPs, silica nanoparticles have been observed to reduce pests when sprayed in infested plants [15].

Longevity has been considered as one of the vital characteristics of an insect’s life cycle due to the role it plays in maintaining population size. Observably, our results show that when the insects fed on food dipped in different concentrations of ZnO NPs, the longevity of adults had a significant difference when compared to insects fed on food dipped in control Table 1. Previous studies recorded that when the insect pests are exposed to insecticides, their longevity is distorted hence the population plummets [22].

Oviposition was greatly reduced due to ingestion of ZnO NPs as compared to the specimen fed on control-treated baby corn. That notwithstanding, the number of eggs laid by the females that fed on baby corn dipped in ZnO were fewer than those who fed on food dipped in control. Moreover, the hatchability of eggs was significantly reduced in all the treatment groups compared to the control [31]. A previous study that exposed *Drosophila* spp. to silver nanoparticles showed that they reduced fertility and vertical climbing in females [32]. Considerably, as a substitute for killing adult insects, it would be beneficial to regulate or manage the insect pest population by plummeting the egg-laying capacity of females and the hatching of the same.

## 5. Conclusions

ZnO NPs demonstrate a promising potential to be used as an alternative to the more lethal insecticides in controlling *S. frugiperda* and other species. Although the environmental effects of using ZnO nanoparticles as an insecticide should be studied further, one obvious benefit of using them as insecticides is the low risk of developing resistance by the insects when used for a long time.

The findings of this study suggest the possibility of exploitation of ZnO nanoparticles not only to control *S. frugiperda* but to significantly reduce their population in the ecosystem through instigation of body deformities, reduced fecundity, reduced oviposition, and reduced hatchability of eggs. It is a valuable tool in integrated pest management regimens.

## Figures and Tables

**Figure 1 insects-12-01017-f001:**
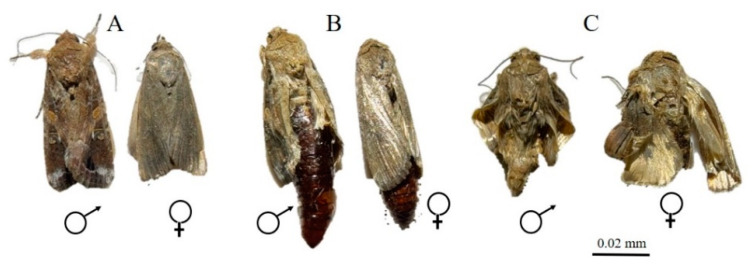
(**A**) Control male and female after emergence; (**B**,**C**) Malformed male and female after emergence.

**Figure 2 insects-12-01017-f002:**
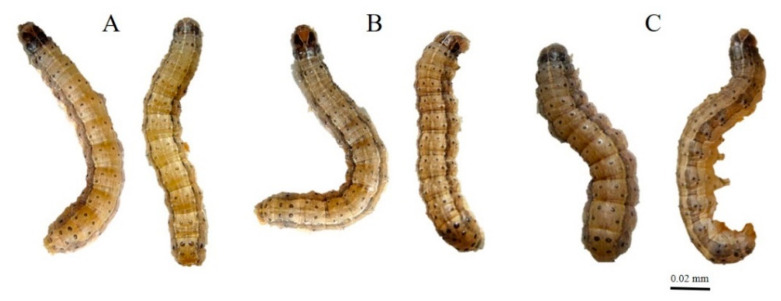
(**A**) Control 5th instar larvae; (**B**,**C**) Malformed 5th instar larvae.

**Figure 3 insects-12-01017-f003:**
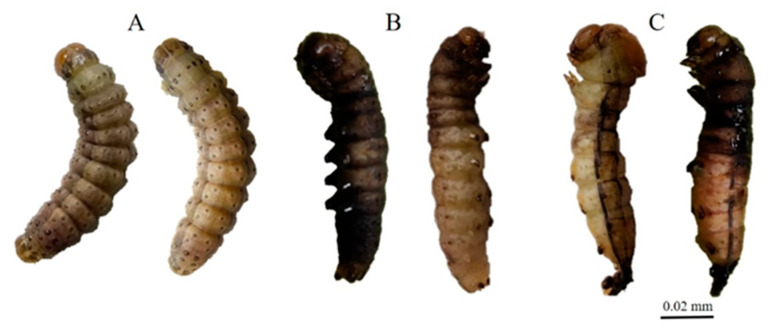
(**A**) Control 6th instar larvae; (**B**,**C**) Malformed larvae 6th instar larvae.

**Figure 4 insects-12-01017-f004:**
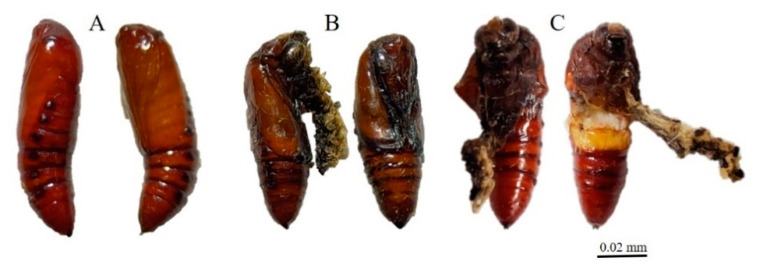
(**A**) Control pupa; (**B**,**C**) Malformed pupa.

**Figure 5 insects-12-01017-f005:**
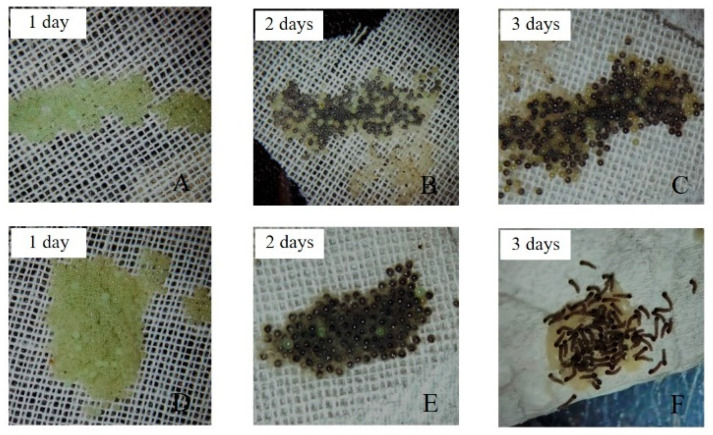
(**A**–**C**) = Eggs from adults fed on food dipped in ZnO NPs; (**D**–**F**) = Eggs from the adult fed on food dipped in control.

**Table 1 insects-12-01017-t001:** Development time and adult longevity (Mean ± SE) of FAW fed on dip food (baby-corn) in ZnO NPs difference concentrations (100–500 ppm).

Parameters	100		200		300		400		500		Control		df	F
	n	Days	n	Days	n	Days	n	Days	n	Days	n	Days		
Egg	80	2.50 ± 0.000	80	2.50 ± 0.000	80	2.50 ± 0.00	80	2.50 ± 0.000	80	2.50 ± 0.000	80	2.50 ± 0.000	-	-
L1	80	2.30 ± 0.105 a	80	2.50 ± 0.115 ab	79	2.60 ± 0.112 ab	80	2.65 ± 0.109 ab	80	2.85 ± 0.082 b	80	2.45 ± 0.114 ab	5	3.100
L2	80	2.15 ± 0.167 a	80	2.25 ± 0.160 a	79	2.30 ± 0.147 a	80	2.35 ± 0.109 a	80	2.55 ± 0.114 a	80	2.00 ±0.126 a	5	1.796
L3	80	1.60 ± 0.112 a	80	1.60 ± 0.112 a	76	1.95 ± 0.114 ab	80	2.35 ± 0.131 bc	80	2.40 ± 0.134 c	80	1.85 ± 0.109 a	5	8.671
L4	73	1.50 ± 0.115 a	77	1.55 ± 0.135 a	73	1.55 ± 0.114 a	76	1.75 ± 0.123 ab	80	1.85 ± 0.820 ab	80	2.05 ± 0.135 b	5	3.292
L5	61	2.25 ± 0.099 ab	72	2.35 ± 0.109 abc	73	2.40 ± 0.169 abc	75	2.60 ± 0.112 bc	80	2.75 ± 0.099 c	80	2.05 ± 0.088 a	5	4.620
L6	39	2.00 ± 0.126 a	34	1.90 ± 0.176 a	34	2.50 ± 0.154 ab	37	2.80 ± 0.172 bc	40	3.30 ± 0.164 c	76	2.20 ± 0.156 a	5	11.241
Prepupa	39	1.50 ± 0.115 a	34	1.55 ± 0.114 a	34	1.60 ± 0.134 a	37	1.65 ± 0.131 a	40	2.10 ± 0.143 b	76	1.45 ± 0.114 a	5	3.505
Pupa	39	9.15 ± 0.167 bc	32	8.45 ± 0.246 b	34	8.95 ± 170 bc	36	9.70 ± 0.219 c	40	7.30 ± 0.317 a	76	9.55 ± 0.114 c	5	16.701
Adults	30	-	20	-	24	-	23	-	26	-	72	-	-	-
Female	22	8.7 ± 0.524 b	11	8.45 ± 0.515 b	16	7.85 ± 0.554 b	14	5.2 ± 0.374 a	17	5.05 ± 0.394 a	39	13.15 ± 0.284 c	5	42.962
Male	8	07.05 ± 0.526 b	9	04.95 ± 0.276 a	8	4.65 ± 2.84 a	9	4.65 ± 2.84 a	9	4.80 ± 0.445 a	33	09.55 ± 0.312 c	5	29.502

Different letters indicate significant differences.

**Table 2 insects-12-01017-t002:** Pupal weight, width, and Length (Mean ± SE) of FAW fed on dip food (baby-corn) in ZnO NPs different concentrations (100–500 ppm) and control.

Treatments (ZnONPs: ppm)	Parameters		
	Weight	Width	Length
100	0.160 ± 0.007 b	0.426 ± 0.008 a	1.397 ± 0.031 a
200	0.130 ± 0.005 a	0.432 ± 0.006 a	1.416 ± 0.025 a
300	0.125 ± 0.005 a	0.422 ± 0.010 a	1.376 ± 0.024 a
400	0.120 ± 0.006 a	0.415 ± 0.007 a	0.415 ± 0.007 a
500	0.113 ± 0.006 a	0.407 ± 0.007 a	1.407 ± 0.029 a
Control	0.186 ± 0.002 c	0.438 ± 0.007 a	1.504 ± 0.015 b
df	5	5	5
F	27.288	2.112	3.457

Different letters indicate significant differences.

**Table 3 insects-12-01017-t003:** Mean (±SE) percentages of pupa and adult normal/abnormal/dead of FAW fed on food (baby-corn) dipped in ZnO NPs different concentrations (100–500 ppm).

Parameters	Treatments (ZnONPs: ppm)						df	F
	100	200	300	400	500	Control		
Pupa								
% Normal	48.75 ± 5.907 a	40.00 ± 4.564 a	42.50 ± 3.227 a	45.00 ± 3.536 a	50.00 ± 2.041 a	95.00 ± 2.887 b	5	28.077
% Abnormal	3.75 ± 1.250 a	5.00 ± 2.041 a	7.50 ± 1.443 a	7.50 ± 1.443 a	6.25 ± 1.250 a	2.50 ± 1.443 a	5	1.846
% Dead	7.50 ± 1.443 ab	10.00 ± 2.041 b	5.00 ± 0.000 ab	8.75 ± 1.250 ab	11.25 ± 2.394 b	2.50 ± 1.443 a	5	4.080
Adults								
% Normal	16.25 ± 2.394 ab	15.00 ± 2.041 ab	17.50 ± 1.443 ab	12.50 ± 1.443 a	21.25 ± 1.250 b	87.50 ± 1.443 c	5	287.506
% Abnormal	21.25 ± 5.154 b	10.00 ± 2.041 ab	12.50 ± 2.500 ab	16.25 ± 3.750 ab	11.25 ± 3.146 ab	2.50 ± 1.443 a	5	3.774

Different letters indicate significant differences.

**Table 4 insects-12-01017-t004:** Adult preoviposition period (APOP), total preoviposition period (TPOP), oviposition period, fecundity, and percent hatching parameters.

Treatments: (ZnONPs: ppm)	Total Females	Reproductive Females	APOP (Days)	TPOP (Days)	Oviposition Period (Days)
100	22	7	2.40 ± 0.152 a	33.55 ± 0.738 a	3.55 ± 0.303 a
200	11	8	2.85 ± 0.167 a	33.45 ± 0.709 a	3.50 ± 0.366 a
300	16	9	2.85 ± 0.167 a	34.55 ± 0.766 a	2.65 ± 0.365 a
400	14	9	2.95 ± 0.185 a	34.00 ± 0.533 a	2.85 ± 0.365 a
500	17	8	3.05 ± 0.185 a	33.20 ± 0.663 a	2.55 ± 0.444 a
Control	39	29	2.60 ± 0.152 a	39.35 ± 0.519 b	5.30 ± 0.145 b
df	-	-	5	5	5
F	-	-	2.035	12.452	8.826

Different letters indicate significant differences.

## Data Availability

Not applicable.

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
