# Peer review of "Insecticidal Effect of Zinc Oxide Nanoparticles against Spodoptera frugiperda under Laboratory Conditions"

_insects, 2021, doi:10.3390/insects12111017_

Round 1

Reviewer 1 Report

Review ID 1413073

Insecticidal Effect of Zinc Oxide Nanoparticles against Spodoptera frugiperda under laboratory conditions in Chiang Mai, Thailand

               A review of works for journals requires a certain introduction. Recently, there is a growing interest in environmentally friendly methods of plant protection. I think the introduction I wrote once fits this manuscript as well.

               For many years, a large number of plant protection products have been reduced. These action was aimed to diminish those preparations that are ineffective and have a negative impact on the environment. In European countries, genetically modified food has not been accepted by consumers. In crop production, there is a lack of effective pesticides to reduce the population of agraphages. This prompted the EU member states to look for other, environmentally friendly methods of plant protection. One of them is the natural defense mechanism of plants based on volatile organic compounds. The release of volatile organic compounds by plants or the production of phytoecdysteroids may become an important element of plant protection in the future.

               This is quite well organized manuscript. I found this “ms” interesting and innovative. However, a few questions must be explained more precisely.

Critical review:

  1.  

„Insecticidal Effect of Zinc Oxide Nanoparticles against Spodoptera frugiperda under laboratory conditions in Chiang Mai, Thailand”.

If this is a lab test then why are you writing about Chiang Mai, Thailand? Does this mean that your lab is different from those in the rest of the globe?

  1. Zinc oxide is not the only solution for modern plant protection. There are more alternatives to the chemical method. Why don't you mention them? The whole Introduction is poorly written.

3.     Materials and Methods as well as Disscussion poorly described. Decide whether you want to write Short Communication or Regular Paper. World literature is abundant.

  1. The rest of the manuscript is reader friendly (Results and Conclussions).

Some other papers to add:

  • Orientation of European corn borer first instar larvae to synthetic green leaf volatiles. Journal of Applied Entomology 137(3): 234-240; 2013.

DOI: 10.1111/J.1439-0418.2012.01719.X

  • Volatile organic compounds released by maize following herbivory or insect extract application and communication between plants. Journal of Applied Entomology 141: 630–643; 2017.

DOI: 10.1111/JEN.12367

  • Sitophilus granarius responses to blends of five groups of cereal kernels and one group of plant volatiles. Journal of Stored Products Research 62: 36-39; 2015.

DOI: 10.1016/J.JSPR.2015.03.007

Author Response

#

ISSUE RAISED

RESPONSE

1

Removing the parts reading Chiang Mai, Thailand from the title due to the fact that this is a laboratory experiment.

„Insecticidal Effect of Zinc Oxide Nanoparticles against Spodoptera frugiperda under laboratory conditions in Chiang Mai, Thailand”.

The title has been revised and the new title reads

“Insecticidal Effect of Zinc Oxide Nanoparticles against Spodoptera frugiperda under laboratory conditions.”

2

Zinc oxide is not the only solution for modern plant protection. There are more alternatives to the chemical method. Why don't you mention them? The whole Introduction is poorly written.

Other nanoparticles have also been talked about in the introduction section as well as the discussion. The introduction was rewritten in a more clear and focused manner.

3

Materials and Methods as well as Disscussion poorly described. Decide whether you want to write Short Communication or Regular Paper. World literature is abundant.

Materials and Methods and the Discussion have been revised for more clarity and valuable deductions.

4

Consideration of three references provided by the Reviewer.

The references have been considered and utilized accordingly.

Reviewer 2 Report

Overall, this paper is a rather pedestrian study of insecticidal efficacy of ZnO nanoparticles, but does document a variety of effects on different life stages and development in design and findings.  The study seems similar to reference [24] without the thiamethoxam and therefore is not unique.  The paper needs some editing for proper English, but is not too badly written.  Specific comments are given below.

Line 45: Deformations without a modifier is pretty ambiguous.   “body deformations” is better

Line 108, “…the region is a plain…”

Line 122:  remove the word ”used”

Line 123:  The link to the webpage for the nanoparticles as given in the Methods is inactive.  I got this message from the provided link:  https://www.dataforthai.com/error404

Line 138:  “deformities” is the word you are looking for.

Tables: 100 ppm as the lowest effective concentration reflects low activity.

Line 157:  Duncan’s multiple range test has fallen out of favor due to its high rate of Type 1 error.  Tukey’s should be used instead, and using this test will probably alter the statistical significance of some comparisons and therefore the conclusions of the study.

Figure 6 shows results already presented in Table 4 and can be removed

Line 322:  What are “very low dosages”?

Lines 354-356:  “Additionally, Zn falls within the micronutrients in the diet of human beings and animals and, therefore, when ingested by humans and animals, they tend to benefit rather than harm them.”  This argument is not appropriate without human exposure data from ZnO nanoparticle treatments in the field.  Too much Zn can be toxic.  This statement should be removed. 

Line 358:  There is no way you will “eradicate” S. frugiperda with these nanoparticles.  Please modify this claim.

Author Response

#

ISSUE RAISED

RESPONSE

1

The study seems similar to reference [24] without the thiamethoxam and therefore is not unique. 

The study follows the methodology of reference (24) as mentioned, however, this study specifically explored the insecticidal effect of ZnO nanoparticles on Spodoptera frugiperda. Also, we discovered some good results on the deformations that was instigated in by the nanoparticle which eventually plays a role in the reduction of population of the insect. The other difference is that the ZnO was not augmented with any other compound.

2

The paper needs some editing for proper English, but is not too badly written.

Editing for English has been improved by a native speaker.

3

Line 45: Deformations without a modifier is pretty ambiguous.   “body deformations” is better

Modified 

4

Line 108, “…the region is a plain…”

Corrected

5

Line 122:  remove the word ”used”

Deleted

6

Line 123:  The link to the webpage for the nanoparticles as given in the Methods is inactive.  I got this message from the provided link:  https://www.dataforthai.com/error404

Apparently, the link is inactive, so we have deleted but the information concerning the characteristics of the Zinc Oxide nanoparticles used is available in the company as mentioned in the text.  

7

Line 138: “deformities” is the word you are looking for.

Thank you, the word was used appropriately.

8

Tables: 100 ppm as the lowest effective concentration reflects low activity.

100ppm which is equivalent to 0.01% means that the effectiveness of the tested ZnO NPs is efficacious. Previous studies e.g. Armstrong et al., 2013 also used similar concentrations and the results corroborate our findings.  

9

Line 157:  Duncan’s multiple range test has fallen out of favor due to its high rate of Type 1 error.  Tukey’s should be used instead, and using this test will probably alter the statistical significance of some comparisons and therefore the conclusions of the study.

The data is Tukey’s has been used

10

Figure 6 shows results already presented in Table 4 and can be removed

Figure 6 and Table 4 are slightly different

11

Line 322:  What are “very low dosages”?

The word ‘very’ has been removed.

12

Lines 354-356: “Additionally, Zn falls within the micronutrients in the diet of human beings and animals and, therefore, when ingested by humans and animals, they tend to benefit rather than harm them.”  This argument is not appropriate without human exposure data from ZnO nanoparticle treatments in the field.  Too much Zn can be toxic.  This statement should be removed.

This statement has been deleted.

13

Line 358:  There is no way you will “eradicate” S. frugiperda with these nanoparticles.  Please modify this claim.

The word ‘eradicate’ has been replaced with the word ‘control’.

Reviewer 3 Report

This manuscript addressed a major issue of managing a global destructive pest, the Fall armyworm using a novel nanaparticle of Zinc Oxide. While the research area is pretty interesting, I raise several major issues that need attention prior to its publication. 

Introduction: This is too lengthy and included several irrelevant sentences that should be removed (eg. lines 52-54).  A simple introduction should start with introducing the pest, their host range, economic damage, currently practiced control strategies and then finish off with the new technology/strategy developed, justifying the research. Unnecessary criticism of the chemical treatments should be avoided when new treatments are not registered!

Materials and Methods: Sections 2.4 doesn't cite a published method, if this is a new method that the authors have developed, they new to clarify this in the beginning! A reference must be given for sex determination of pupae (line 140). Criteria for all the parameters (eg. duration of instars, deformation, longevity) studied (line 141) need to be expanded and given in full.

Results: This section need serious revision to match the Biological Traits described in the Methods section, it seems to be vaguely written!

Figures: Photos provided in the figures need to have clarity and should have white background to be able to see the deformities etc. In Figure 2, I can't see much malformation in the treated larvae!

Discussion: It should start with the aim of the current study and highlight the findings. The authors have again discussed the topic same way as the Introduction. The results should be discussed in relation to earlier published data and critically analyse their implication towards the management of the pest.

Conclusion: This section should highlight the importance of the current finding and suggests ways, how the laboratory study can be taken to the field. 

Author Response

#

ISSUE RAISED

RESPONSE

1

Introduction: This is too lengthy and included several irrelevant sentences that should be removed (eg. lines 52-54).  A simple introduction should start with introducing the pest, their host range, economic damage, currently practiced control strategies and then finish off with the new technology/strategy developed, justifying the research. Unnecessary criticism of the chemical treatments should be avoided when new treatments are not registered!

The introduction has been rewritten and made precise and more focused. The suggested flow of information has been considered and adhered to.

2

Materials and Methods: Sections 2.4 doesn't cite a published method, if this is a new method that the authors have developed, they new to clarify this in the beginning! A reference must be given for sex determination of pupae (line 140). Criteria for all the parameters (eg. duration of instars, deformation, longevity) studied (line 141) need to be expanded and given in full.

The sexual determination of the pupa was done by observing the position of the ovipositor under Zeiss Stemi 508 Compound Microscope according to procedure used by Zhang et al., 2017. Also, the criteria for the duration of instars, deformation and longevity has been expounded.

3

Results: This section need serious revision to match the Biological Traits described in the Methods section, it seems to be vaguely written!

The materials and methods have been revised for clarity as well as the results section for synchrony.

4

Figures: Photos provided in the figures need to have clarity and should have white background to be able to see the deformities etc. In Figure 2, I can't see much malformation in the treated larvae!

The photos have been taken again on a white background and are clearer. For the case of Figure 2, although, it is difficult to tell of the deformations on larvae in a picture, but when compared with the control, you can see the difference both in the picture and physically.

5

Discussion: It should start with the aim of the current study and highlight the findings. The authors have again discussed the topic same way as the Introduction. The results should be discussed in relation to earlier published data and critically analyse their implication towards the management of the pest.

The discussion has been revised and the aim of the study mentioned at the onset. We have looked critically at the previous data and used it to support our results as suggested.

6

Conclusion

The conclusion section has been revised and the importance of the study made clearer

Round 2

Reviewer 2 Report

Response to the author's comment:  100 ppm as a treatment is A LOT of material.  If you doubt this, check to see what final concentrations are used for commercial products.  A lot of insecticides are active at ppb levels.  The fact that another paper used the same high concentration supports that it has activity at this high level, but does not mitigate the argument that 100 ppm is a high concentration.

My only additional concern is that the information on the nanoparticles used still references the same company, but now does not include even an inactive URL.  I can a link for this company online, but no link to product information like the authors allege.  In any event, information on the nanoparticles used should be available to the authors if they purchased them, or alternatively, if that info is lost, this sentence referencing the Company in question should be removed.  Of course, the lack of this information makes replicating this study problematic in the future, although it does not preclude publication.

Figure 6 is still redundant.

Author Response

1

Response to the author's comment:  100 ppm as a treatment is A LOT of material.  If you doubt this, check to see what final concentrations are used for commercial products.  A lot of insecticides are active at ppb levels.  The fact that another paper used the same high concentration supports that it has activity at this high level, but does not mitigate the argument that 100 ppm is a high concentration.

Your comment is noted. Indeed, 100 ppm is a lot and we hope to do more experiments in the future to try to find a lower concentration either through augmentation with entomopathogenic fungi or other biological control measures. However, the beauty about ZnO nanoparticles, is that they are safer for the environment as compared to other chemicals.

2

My only additional concern is that the information on the nanoparticles used still references the same company, but now does not include even an inactive URL.  I can a link for this company online, but no link to product information like the authors allege.  In any event, information on the nanoparticles used should be available to the authors if they purchased them, or alternatively, if that info is lost, this sentence referencing the Company in question should be removed.  Of course, the lack of this information makes replicating this study problematic in the future, although it does not preclude publication.

Thank for your comment, fortunately we managed to find a link to the organization and it is provided in the manuscript. The link is http://www.nano.kmitl.ac.th

3

Figure 6 shows results already presented in Table 4 and can be removed

Figure 6 has been deleted.  

Reviewer 3 Report

Accept

Author Response

No comment required a response. 

We appreciate your support and recommendations in the first review.